# Textural and Functional Properties of Skimmed and Whole Milk Fermented by Novel *Lactiplantibacillus plantarum* AG10 Strain Isolated from Silage

Elena Nikitina [1,2,*], Tatyana Petrova [1], Adel Vafina [1], Asya Ezhkova [3], Monyr Nait Yahia [2] and Airat Kayumov [2,3,*]

[1]  Department of Meat and Milk Technology, Faculty of Food Technology, Kazan National Research Technological University, 420015 Kazan, Russia; tanya.yurtaeva.95@mail.ru (T.P.); adelia88@mail.ru (A.V.)
[2]  Department of Genetics, Institute of Fundamental Medicine and Biology, Kazan (Volga Region) Federal University, 420008 Kazan, Russia; monyraniir@gmail.com
[3]  Department of Physiology and Patophysiology, Kazan State Academy of Veterinary Medicine Named after N.E. Bauman, 420029 Kazan, Russia; egkova-am@mail.ru
*  Correspondence: ev-nikitina@inbox.ru (E.N.); kairatr@yandex.ru (A.K.)

**Abstract:** Milk fermentation by lactic acid bacteria both enhances its nutritional value and provides probiotic strains to correct the intestinal microflora. Here, we show the comparative analysis of milk fermented with the new strain, *Lactiplantibacillus plantarum* AG10, isolated from silage and the industrial strain *Lactobacillus delbrukii* subs. *bulgaricus*. While the milk acidification during fermentation with *L. plantarum* AG10 was lower compared with *L. bulgaricus*, milk fermented with *L. plantarum* AG10 after a 14-day storage period retained a high level of viable cells and was characterized by an increased content of exopolysaccharides and higher viscosity. The increased EPS production led to clot formation with higher density on microphotographs and increased firmness and cohesiveness of the product compared with *L. bulgaricus*-fermented milk. Furthermore, the *L. plantarum* AG10-fermented milk exhibited increased radical-scavenging activity assuming lower fat oxidation during storage. Taken together, these data suggest that *L. plantarum* AG10 seems to be a promising starter culture for dairy products with lowered levels of lactic acid, which is important for people with increased gastric acid formation.

**Keywords:** *Lactiplantibacillus plantarum*; antioxidant properties; starter cultures; probiotics; fermented milk; texture





## 1. Introduction

The health benefits of dairy products governed by the biologically active components produced by Lactic acid bacteria (LAB) during milk fermentation have been known since medieval times [1]. Milk fermentation with LAB leads to improvements in the texture, flavor, bioavailability, and nutritional value of many dairy products [2,3]. Thus, the transformation of lactose into lactic acid causes the coagulation of milk proteins, mainly casein, while other metabolites, such as diacetyl, acetoin, etc., provide a specific aroma and flavor for the product [4,5]. The lactic acid produced by LAB during milk fermentation leads to a decrease in pH to 4.0–4.5 and increase in titratable acidity, and serves as the main preservative of the product. The low pH inhibits the growth of pathogenic microorganisms causing food spoilage and poisoning [6–8] and is responsible for antifungal properties [9–14].

Besides the modification of the product structure and organoleptic properties, LAB also act directly to prevent some severe disorders, from the dysbiosis of microbiota to allergy, asthma, and cancer [15–19]. The mechanism of the functional role of fermented dairy products is achieved either directly through the interaction of the gut microbiota with consumed probiotic microorganisms (probiotic effect) or indirectly via microbial metabolites, such as vitamins, peptides, oligosaccharides, and organic acids, generated during the

fermentation process (biogenic effect) [1,20–22]. The proteolytic degradation of casein leads to the release of short peptides, some of which have antimicrobial, antithrombotic, and immunomodulatory properties [23–27], as well as antioxidants [28]. Furthermore, bacterial enzymes transform milk carbohydrates into oligosaccharides, some of which have prebiotic properties [29]. LAB strains also vary in their ability to produce exopolysaccharides (EPS), which have been reported as antioxidant agents [30], exhibit immune-regulating properties [31], and provide the texture [32] of dairy products.

Moreover, since many LAB have been shown to hydrolyze fats and cholesterol in vitro and in animal studies [33,34], in the long-term perspective, LAB could be a tool to correct the blood cholesterol levels and reduce the risk of cardiovascular disease and obesity.

To date, various lactic acid bacteria, such as *Streptococcus thermophilus*, *L. lactis*, *L. helveticus*, and *L. delbrueckii* subsp. *bulgaricus*, are used as starter strains to produce various fermented dairy products. Since milk fermentation with various LAB strains leads to the formation of final products with different properties, the proper selection and balance of lactic acid bacteria used as a starter culture are crucial to obtain a product with a desirable texture and flavor [35–37]. While *S. thermophilus* and *L. bulgaricus* are the two main bacteria used for yogurt production, minor strains, which include non-specialized lactic acid bacteria (NSLAB) also capable of milk fermentation and acting as probiotics, are added along with the main starter cultures to obtain products with different properties [38]. Most of the LAB strains recommended as probiotics are from the family *Lactobacillaceae* [39] and could be isolated from various sources, including feces, fermented milk, fish, and vegetables, such as silage [40,41] and rarer sources, such as fermented olives of Cv. conservolea and Halkidiki [42].

Here, we show a comparative investigation of milk fermentation by the new heterofermentative *L. plantarum* AG10 strain isolated from silage [43] and homofermentative industrial strain *Lactobacillus delbrukii* subs. *bulgaricus*. We demonstrate the differences in various important properties of both skimmed and whole milk fermented with these two strains. Along with high in vitro tolerance to low pH, bile salts, antibacterial, and probiotic properties discussed earlier, *L. plantarum* AG10 seems to be a promising strain as a minor component of starter cultures for dairy products.

## 2. Materials and Methods

### 2.1. Strains and Milk Fermentation

*Lactiplantibacillus* (*Lactobacillus*) *plantarum* AG10 was isolated from silage, provided a high milk acidification rate, and exhibited potential probiotic properties [43]. *Lactobacillus delbrueckii* subs. *bulgaricus* ("Lactosynthesis", Moscow, Russia) served as a reference strain. LABs were stored in de Man, Rogosa, and Sharpe (MRS) broth (Himedia, India) with 50% glycerol at −80 °C. Bacteria were seeded from the stock into MRS broth, grown at 37 °C for 24 h, and then inoculated into skimmed milk to obtain the pre-culture.

The pre-cultures of lactic acid bacteria (LAB) were prepared by incubation at 40 °C for 16 h in skimmed UHT milk (Valio®, Russia) and then inoculated (5 % *v/v*) into the skimmed or whole milk (0.05 or 3.2 % fat Valio®, Russia). The fermentation was carried out at 40 °C for 8 h, as milk fermentation is generally performed for yogurt production under industrial conditions. Thereafter, the fermented samples were incubated at 6 °C for 16 h for maturation and then stored at 4–6 °C. The physicochemical, textural, and antioxidant properties of the fermented milk were evaluated after 24 h, 7, and 14 days of storage.

*Escherichia coli* MG1655 (K-12) and *Staphylococcus aureus* subsp. *aureus* ATCC 29213 were stored with 50% glycerol at −80 °C and grown on LB–agar plates as test bacteria when evaluating the antibacterial activity of the LAB.

### 2.2. Preparing of Protein-Free Extract (PFE)

Five milliliters of 1% trichloroacetic acid solution were added to five grams of fermented milk and mixed. After 5 min of incubation at 25 °C, the precipitate was removed by centrifugation for 15 min at $10,000 \times g$. The supernatant was used as a PFE for fer-

ric reducing antioxidant power (FRAP) analysis and radical scavenging ability (RSA), as described below.

### 2.3. Quantitative Chemical Analysis of Fermented Skimmed and Whole Milk

The quantitative analysis of proteins, fat, dry matter, and density in the fermented milk was performed using Infra LUM® FT-12 equipment (Lumex, Russia) with the software and calibration data recommended for the product "yogurt". The total carbohydrates and salt in whey were determined by using "Klever-M" equipment (Biomer, Russia) in the supernatant after fermented milk centrifugation at $3000 \times g$ for 15 min. The dextrose content of whey was determined by Accu-Chek active GC (Roche, Germany).

To determine the total titratable acidity (TTA), 10 g of fermented milk was suspended in 20 mL of pure water and titrated with 0.1 M NaOH to a final pH of 8.2, detected by 1% phenolphthalein in ethanol. The TTA of fermented milk samples was expressed in Thorner degrees (°T).

To determine the water-holding capacity (WHC), 20 g of fermented milk was stored at +6 °C for 24 h (Y) and centrifuged for 10 min at 3000 rpm. The released whey (W) was removed and weighed. The water-holding capacity (WHC) of fermented milk was calculated as WHC = $(Y - W)/Y \times 100\%$.

To measure the syneresis, the fermented and cooled milk was centrifuged at 1000 rpm for 5 min. The whey (W) was removed and weighed. The syneresis (Syn) of fermented milk was calculated as Syn = $W/Y \times 100\%$.

### 2.4. Antibacterial Activity

The antibacterial activity of the fermented milk and its fractions was evaluated using the spot-on-lawn approach [44]. The inhibition values were ordered by a ranking system of five levels of activity corresponding to very high (>7 mm radius), high (>4 mm radius), moderate (>2 mm radius), low (>1 mm radius), and without significant activity (<1 mm) [45]. Each test was performed in triplicate.

### 2.5. Determination of the Total Amount of Exopolysaccharides (EPS)

The isolation and quantification of exopolysaccharides were carried out as described by Feldmanel et al. [46], with modifications. Briefly, 10 g of the fermented milk was incubated in a flask at 100 °C for 30 min. After cooling until 4 °C, the samples were centrifuged at 4000 rpm for 30 min, and 0.14 mL of 85% trichloroacetic acid was added to 4 mL of the supernatant. After 5 min of incubation, the precipitate was removed by centrifugation at 8000 rpm for 10 min. To precipitate the EPS, 1 mL of supernatant was mixed with 3 mL of chilled (−20 °C) ethanol and incubated for 48 h at 4 °C. The EPS was harvested by centrifugation for 10 min at 8000 rpm, and the precipitate was dissolved in 10 mL of pure water.

To quantify exopolysaccharides, 400 μL of the sample was mixed with 400 μL of a fresh 5% phenol solution in water, and 2 mL of 96% sulfuric acid was added. Samples were incubated at 30 °C for 10 min, and then stirred and left for 10 min. The absorbance was measured at 490 nm, and the reaction mixture with pure water added instead of the sample served as a reference. The EPS was calculated by using a calibration curve measured with glucose solutions and presented in milligrams of dextrose per gram of sample.

### 2.6. Apparent Viscosity

The apparent viscosity of the fermented milk samples was measured using a viscometer (RV-DVIII, Brookfield programming Rheometry, Inc., China). Spindle #3 and a rate of 30 $\text{min}^{-1}$ were selected in configuration to provide torque in the range of 10% to 90% during measurement, as suggested by the manufacturer. The temperature of the samples was maintained at 6 °C ± 1 °C during the test. Measurements were carried out in 3 replicates for each treatment and results were expressed in mPa.s.

### 2.7. Textural Studies

The texture profile analysis [47] test was carried out using an ST-2 texture analyzer (Quality Laboratory JSC, Moscow, Russia) with a cylindrical probe of 36 mm in diameter and 35 mm in height, which penetrated the undisturbed samples of fermented milk. Two cycles were applied to a depth of 10 mm at the rate of 0.5 mm s$^{-1}$, touch force 7 g. Fermented milk was tested in a chemical beaker with a diameter of 50 mm, and the height of the sample was 25 mm. As a result, a plot of force versus time was obtained for each sample using the software ST-2 for Windows (Quality Laboratory JSC, Moscow, Russia). The following factors were determined: Firmness (g), Fracturability, Elasticity, Adhesion force (g), Adhesiveness, Cohesiveness, Gumminess, Springiness, and Chewiness [48].

### 2.8. Scanning Electron Microscopy

The microstructure of the fermented milk samples after 7 days of storage was evaluated by SEM. Briefly, the samples were fixed with 2.5% glutaraldehyde for 4–5 h; subsequently, they were washed three times with 0.2 M Na–K phosphate buffer (pH 7.0) and then dehydrated by 30%, 40 %, 50%, 60 %, 70%, and 80% (twice for each concentration) for 15 min, followed by 95% ethanol dehydration three times for 30 min. The samples were mounted on metal stubs and coated with a gold–palladium alloy (∼10 nm thickness) using a Quorum Q150T ES coating machine. Samples were observed using a self-emission scanning electron microscope Merlin (Carl Zeiss; Germany) at an acceleration voltage of 5 kV, with a secondary electrons detector and magnification of 10,000×.

### 2.9. Ferric Reducing Antioxidant Power Assay (FRAP)

The ferric reducing antioxidant power (FRAP) assay was carried out following the procedure described in [49] with modifications. Fermented milk and whey were pre-diluted 2-fold; PFE was used in its initial form. Briefly, 1 mL of the sample was mixed with 1 mL of 0.2 M potassium sodium phosphate buffer (pH 6.5) and 1 mL of 1% potassium ferricyanide. The reaction mixture was incubated for 20 min at 50 °C, cooled, and 1 mL of 10% trichloroacetic acid was added. The mixture was centrifuged at 2000 rpm for 10 min at 25 °C; the supernatant was diluted twice with pure water (2 mL + 2 mL) and 400 μL of 0.1% FeCl$_3$ was added. For the reference, a buffer was added instead of the potassium ferricyanide. The absorbance was measured at 700 nm on a spectrophotometer SF-2000 (Spectr, Saint-Petersburg, Russia) and the reducing force was expressed as the absorbance difference between the samples and reference.

### 2.10. Evaluation of Radical-Scavenging Ability (RSA) by 2,2-Di-phenyl-1-picrylhydrazyl (DPPH) Assay

The radical-scavenging capacity was analyzed according to [50] with modifications. Briefly, 1 mL of sample (5-fold-diluted fermented milk, 5-fold-diluted whey, or 2.5-fold-diluted PFE) was mixed with 1 mL of freshly prepared DPPH solution (0.12 mM in ethanol) and incubated at 25 °C in the dark for 30 min. The reaction mixture was centrifuged for 2 min at 10,000 rpm and the absorbance was read at 517 nm on a spectrophotometer SF-2000 (Russia). As a reference, the DPPH solution (0.12 mM in ethanol) was used. The radical-scavenging activity was calculated as: DPPH x scavenging activity % = [(control absorbance − extract absorbance)/(control absorbance)] × 100 %.

### 2.11. Evaluation of Peroxide Value of Fat

The peroxide value was determined as described in [51]. The whole fermented milk samples (3 g) were weighed in a 100 mL glass Erlenmeyer flask. Then, the sample was incubated for 3 min at 60 °C in a water bath to melt the fat, 30 mL of acetic acid–chloroform solution (3:2 *v/v*) was added, and the mixture was agitated for 3 min to dissolve the fat. The milk particles were removed by filtration using Whatman filter paper N1. Saturated potassium iodide solution (0.5 mL) and starch solution were added to the filtrate. The titration was continued by 0.01 N sodium thiosulfate solution. POV was expressed in

milli-equivalent peroxide per kilogram of sample: POV (meq/kg) = ((S × 0.01)/W) × 100, where "S" is the volume of titration (mL) and "W" is the sample weight (g).

### 2.12. Evaluation of Thiobarbituric Acid-Reactive Substance (TBARS)

The 2-thiobarbituric acid (TBA) values were determined as described in [52]. The milk samples (2 g) were blended with 10 mL of 20% trichloroacetic acid solution (200 g/L of trichloroacetic acid in 135 mL/L phosphoric acid solution) in a homogenizer for 30 s. The milk particles were removed by filtration through Whatman filter paper N4. Then. 2 mL of 0.02 M aqueous TBA solution (3 g/L) was added to 2 mL of filtrate and incubated at 100 °C for 30 min. After cooling, the absorbance of the supernatant was measured at 532 nm using an SF-2000 spectrophotometer (Russia). The TBA values were calculated from a standard curve and expressed in milligrams of malonaldehyde per kilogram (MA/kg) of sample.

### 2.13. Evaluation of Free Fatty Acid Value

The free fatty acid value was determined according to the AOCS Official Method [53]. The sample (5 g) was dissolved in 30 mL of chloroform using a homogenizer at 10,000 rpm for 1 min. The milk particles were removed by filtration through Whatman filter paper N1. After the addition of five drops of phenolphthalein (1% solution in EtOH) as an indicator, the titration was performed with 0.01 M potassium hydroxide solution. The FFA value was calculated as follows: FFA (%) = (mL titration × Molarity of KOH × 28.2)/g of sample.

### 2.14. CFUs Count

To evaluate the number of viable cells in fermented milk, CFUs were counted using the drop-plate assay [54] with modifications [55]. Briefly, a 10-fold dilution series in sterile PBS was prepared and 5 μL from each dilution was plated onto MRS agar followed by incubation at 37 °C for 24 h. CFUs were counted from the two last drops, typically containing 5–15 colonies. Data from 5 independent experiments were expressed as medians with the interquartile range (IQR).

### 2.15. Statistical Analysis

All experiments were carried out in triplicate. Five replicates were performed for DPPH analysis. The results were analyzed with the non-parametric Kruskal–Wallis test using GraphPad Prism, version 6.0, San Diego, CA, USA. Differences were considered significant at $p < 0.05$.

## 3. Results

### 3.1. Characteristics of Fermented Milk

The skimmed and whole milk were fermented for 8 h at 40 °C by either L. plantarum AG10 or L. bulgaricus as a reference strain and stabilized at 6 °C for 16 h, followed by storage at 4 °C. For both samples, various technological and physicochemical parameters, as well as antimicrobial activity, were determined (Table 1).

Both skimmed and whole milk fermented by *L. plantarum* AG10 were characterized by a higher pH value and lower TTA in comparison with milk fermented by *L. bulgaricus* (TTA ~90 and 130, and ~95 and ~150 T, respectively), suggesting lower lactic acid production by *L. plantarum* AG10 and, subsequently, a lower acidification rate. The total amount of protein after fermentation and during storage did not differ significantly between strains for both milk types. In contrast, the level of carbohydrates in *L. plantarum*-fermented skimmed milk was 10% lower compared with the *L. bulgaricus* product. Additionally, the amount of dextrose, one of the lactose cleavage metabolites, decreased from 5.6 ± 0.28 to 4.5 ± 0.59 mmol/l after 14 days of storage in samples obtained with *L. plantarum*, suggesting saccharide consumption by this LAB strain. Similar dependencies were observed in the whole fermented milk, although at a lower absolute value. Additionally, the dry matter content and density of products were insignificantly higher in the milk fermented with *L. bulgaricus*.

**Table 1.** Technological and physicochemical parameters of non-fat milk and whole milk fermented with *L. bulgaricus* or *L. plantarum* AG10 strains after 1, 7, and 14 days of storage at 4 °C. Asterisks indicate a significant difference at $p < 0.05$ between strains. Mean values $\pm$ SD are shown (*n* = 3).

| Strains | | *L. bulgaricus* | | | *L. plantarum* AG10 | | |
|---|---|---|---|---|---|---|---|
| **Days** | | 1 | 7 | 14 | 1 | 7 | 14 |
| Skimmed Milk (fat = 0.05%) | | | | | | | |
| pH | | 4.5 ± 0.3 | 4.5 ± 0.1 | 4.4 ± 0.2 | 5 ± 0.1 * | 5.1 ± 0.2 * | 5.1 ± 0.1 * |
| Total titratable acidity, T° | | 122 ± 8 | 130 ± 6 | 136 ± 4 | 87 ± 4 * | 91 ± 2 * | 93 ± 9 * |
| Total proteins, % | | 3.8 ± 0.6 | 3.9 ± 0.4 | 3.7 ± 0.5 | 3.8 ± 0.1 | 3.8 ± 0.1 | 3.8 ± 0.1 |
| Total carbohydrates, % | | 4.6 ± 0.1 | 4.5 ± 0.1 | 4.5 ± 0.2 | 4.2 ± 0.1 * | 4.2 ± 0.1 * | 4.0 ± 0.3 * |
| Dextrose, mmol/l of whey | | 5.3 ± 0.1 | 5.4 ± 0.3 | 5.2 ± 0.6 | 5.2 ± 0.1 | 5.6 ± 0.3 | 4.5 ± 0.6 * |
| Salts, % | | 0.73 ± 0,03 | 0.73 ± 0.02 | 0.74 ± 0.03 | 0.63 ± 0.04 | 0.68 ± 0.01 | 0.64 ± 0.21 |
| Density, kg/m$^3$ | | 1037.2 ± 0.6 | 1037.6 ± 1.4 | 1037.0 ± 1.3 | 1035.4 ± 1.5 | 1035.8 ± 0.2 | 1038.9 ± 1.1 |
| Dry matter, % | | 9.54 ± 0.50 | 9.59 ± 0.29 | 9.42 ± 0.35 | 8.95 ± 0.22 | 9.07 ± 0.01 | 8.94 ± 0.42 |
| Antimicrobial activity * | *E. coli* | ++ | ++ | ++ | ++ | ++ | + |
| | *S. aureus* | ++ | ++ | ++ | + | + | + |
| Whole Milk (fat = 3.2 %) | | | | | | | |
| pH | | 4.2 ± 0.1 | 4.1 ± 0.1 | 4.1 ± 0.1 | 4.8 ± 0.1 * | 4.8 ± 0.1 * | 4.9 ± 0.1 * |
| Total titratable acidity, T° | | 143 ± 67 | 152 ± 5 | 155 ± 6 | 92 ± 7 * | 95 ± 6 * | 99 ± 3.1 * |
| Total proteins, % | | 3.7 ± 0.1 | 3.6 ± 0.0 | 3.5 ± 0.1 | 3.5 ± 0.1 | 3.5 ± 0.1 | 3.4 ± 0.1 |
| Total carbohydrates, % | | 4.0 ± 0.1 | 4.1 ± 0.1 | 4.1 ± 0.0 | 3.9 ± 0.1 | 3.9 ± 0.1 | 3.9 ± 0.0 |
| Dextrose, mmol/l of whey | | 2.3 ± 0.3 | 2.2 ± 0.4 | 2.1 ± 0.2 | 1.8 ± 0.1 * | 1.7 ± 0.2 * | 1.6 ± 0.1 * |
| Fat, % | | 3.2 ± 0.3 | 3 ± 0.3 | 2.6 ± 0.3 | 2.7 ± 0.1 * | 2.2 ± 0.2 * | 2 ± 0.1 * |
| Salts, % | | 0.68 ± 0.04 | 0.63 ± 0.04 | 0.63 ± 0.01 | 0.68 ± 0.02 | 0.65 ± 0.01 | 0.64 ± 0.01 |
| Density, kg/m$^3$ | | 1031 ± 0.1 | 1032 ± 0.6 | 1032 ± 1.1 | 1031 ± 1.0 | 1031 ± 0.5 | 1031 ± 0.4 |
| Dry matter, % | | 11.69 ± 0.22 | 11.35 ± 0.13 | 11.05 ± 0.30 | 10.86 ± 0.60 | 10.04 ± 0.09 | 10.09 ± 0.12 |
| Antimicrobial activity [1] | *E. coli* | ++ | ++ | ++ | ++ | ++ | + |
| | *S. aureus* | ++ | ++ | ++ | ++ | ++ | + |

[1] The antimicrobial activity is evaluated by pathogens growth repression zone: + less than 3 mm, ++ more than 3 mm.

Finally, the antagonistic activity of LAB against E. coli and S. aureus was evaluated (Table 1). The repression of pathogen growth by *L. plantarum* AG10 on agar plates with 1% glucose was lower than that of *L. bulgaricus*, which may have been because of lower acid production.

### 3.2. The Exopolysaccharides Production

The consistency and structure of fermented milk mainly depend on the presence of highly viscous polysaccharides. Therefore, the ability to produce exopolysaccharides (EPS) is one of the important features of starter cultures. After 14 days of storage, the EPS content in the skimmed milk fermented by *L. plantarum* AG10 was 1.4-fold higher in comparison with that of *L. bulgaricus*-fermented milk, as judged by dextrose measurement (Figure 1). In the case of whole milk, significant EPS accumulation by *L. plantarum* AG10 already occurred after 7 days of storage.

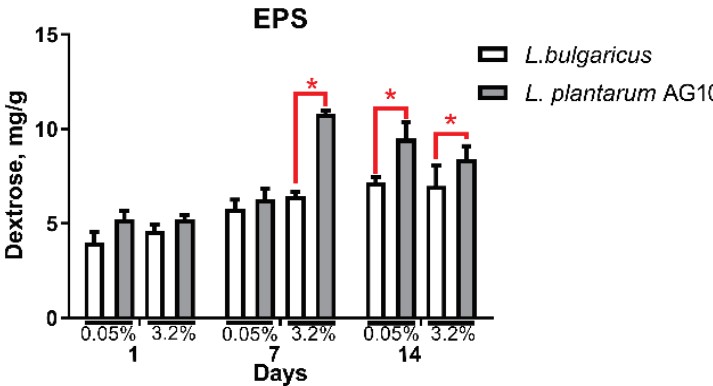

**Figure 1.** The amount of EPS in skimmed and whole milk fermented with either L. *bulgaricus* (shown in white) or *L. plantarum* AG10 (shown in gray) as judged by dextrose measurement after 1, 7, and 14 days of storage. Asterisks indicate a significant difference at $p < 0.05$. The initial fat percentage in the milk is shown under the *x*-axis.

*3.3. Textural Properties of Fermented Milk*

As a consequence of higher EPS production, *L. plantarum* AG10-derived skimmed samples exhibited a significantly higher viscosity index, while syneresis and WHC did not differ up to the 14th day of storage. The fermented whole milk had 1.5-fold lower syneresis on the 14th day of storage in comparison with samples fermented by *L. bulgaricus*, while the WHC of both samples did not differ (Figure 2).

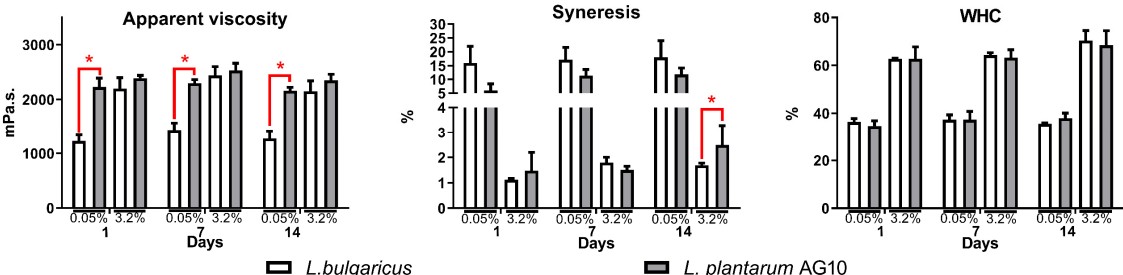

**Figure 2.** Textural characteristics of skimmed and whole milk fermented with either *L. bulgaricus* (shown in white) or *L. plantarum* AG10 (shown in gray) after 1, 7, and 14 days of storage. Asterisks show a significant difference at $p < 0.05$. The % of initial fat in the milk is shown under the *X*-axis.

Some other textural properties observed on the 14th day for the fermented milk significantly depended on the microorganism being more favorable in *L. plantarum* AG10 fermented milk (Table 2). Thus, in the case of both skimmed and whole milk, the adhesiveness of the product produced by *L. plantarum* AG10 was higher compared with that of the *L. bulgaricus*-derived product. Additionally, the milk fermented by *L. plantarum* AG10 demonstrated a greater ability for cohesion, suggesting higher resistance to destruction. Furthermore, the milk fermented by *L. plantarum* AG10 was characterized by two-fold lower springiness, and higher gumminess and chewiness. No such difference was observed in the case of whole milk, while the tendency for the chewiness index to increase remained in the case of the *L. bulgaricus*-derived samples.

**Table 2.** Textural profile of skimmed and whole milk fermented with either *L. bulgaricus* (shown in white) or *L. plantarum* AG10 (shown in gray) after 1, 7, and 14 days of storage. Asterisks show a significant difference at $p < 0.05$. Mean values $\pm$ SD are shown ($n$ = 3).

| Storage | Firmness, g | Fracturability | Elasticity | Adhesion Force, g | Adhesiveness | Cohesiveness | Gumminess | Springiness, s | Chewiness |
|---|---|---|---|---|---|---|---|---|---|
| | | | *L. bulgaricus*, (fat = 0.05%) | | | | | | |
| 1 | 32.6 ± 1.0 | 6.35 ± 0.02 | 0.50 ± 0.01 | 6.80 ± 0.04 | 3.13 ± 1.20 | 0.27 ± 0.00 | 8.8 ± 0.3 | 20.8 ± 1.1 | 164.2 ± 6.5 |
| 7 | 32.5 ± 1.1 | 5.86 ± 0.03 | 0.57 ± 0.01 | 6.30 ± 0.10 | 2.98 ± 0.91 | 0.25 ± 0.02 | 9.2 ± 1.1 | 19.8 ± 0.9 | 185.1 ± 4.5 |
| 14 | 32.5 ± 1.0 | 5.96 ± 0.03 | 0.57 ± 0.02 | 6.10 ± 0.21 | 4.32 ± 1.03 | 0.32 ± 0.01 | 10.5 ± 0.9 | 19.7 ± 0.2 | 208.7 ± 35.2 |
| | | | *L. plantarum* AG10, (fat = 0.05%) | | | | | | |
| 1 | 30.2 ± 1.1 | 4.46 ± 0.02 | 0.69 ± 0.03 | 5.80 ± 0.22 | 4.32 ± 1.31 | 0.30 ± 0.003 | 17.7 ± 0.1 * | 9 ± 0.3 * | 160.4 ± 6.3 |
| 7 | 31.7 ± 1.1 | 5.84 ± 0.02 | 0.75 ± 0.01 | 5.90 ± 0.1 | 4.46 ± 0.64 | 0.33 ± 0.01 * | 18.5 ± 0.5 * | 10.3 ± 0.6 * | 191.2 ± 2.5 * |
| 14 | 32.3 ± 1.0 | 5.72 ± 0.05 | 0.62 ± 0.01 | 6.10 ± 0.02 | 4.87 ± 0.52 | 0.38 ± 0.03 * | 19.3 ± 1.2 * | 12.2 ± 1.2 * | 2365.8 ± 8.5 * |
| | | | *L. bulgaricus*, (fat = 3.2%) | | | | | | |
| 1 | 39.2 ± 1.2 | 11.92 ± 0.01 | 0.22 ± 0.02 | 10.00 ± 0.01 | 3.70 ± 2.01 | 0.25 ± 0.01 | 20 ± 2.1 | 9.6 ± 0.6 | 194.1 ± 2.3 |
| 7 | 51 ± 1.4 | 16.29 ± 0.01 | 0.14 ± 0.02 | 13.80 ± 0.10 | 4.77 ± 0.51 | 0.20 ± 0.01 | 27.5 ± 1.2 | 10.2 ± 0.2 | 281.4 ± 6.5 |
| 14 | 54.7 ± 1.4 | 19.69 ± 0.03 | 0.12 ± 0.04 | 14.40 ± 0.05 | 4.82 ± 0.86 | 0.20 ± 0.01 | 27.8 ± 0.9 | 11.1 ± 0.9 | 309.9 ± 10.5 |
| | | | *L. plantarum* AG10, (fat = 3.2%) | | | | | | |
| 1 | 38.3 ± 1.0 | 10.28 ± 0.02 | 0.30 ± 0.03 | 8.70 ± 0.03 | 4.14 ± 1.26 | 0.28 ± 0.01 | 18.9 ± 1.7 | 10.8 ± 1.1 | 205.1 ± 6.8 * |
| 7 | 52.4 ± 1.0 | 17.18 ± 0.02 | 0.13 ± 0.01 | 13.30 ± 0.02 | 5.35 ± 0.68 | 0.23 ± 0.01 | 26.6 ± 0.9 | 12.2 ± 1.0 | 324.5 ± 2.5 * |
| 14 | 59 ± 1.1 | 22,07 ± 0.04 | 0.10 ± 0.01 | 15.80 ± 0.02 | 6.13 ± 1.11 | 0.22 ± 0.01 | 27.8 ± 1.1 | 13.2 ± 0.3 | 368.8 ± 10.4 * |

### 3.4. Scanning Electron Microscopy

The ultrastructures of the milk fermented by *L. bulgaricus* and *L. plantarum* AG10 were further analyzed by SEM. In the case of *L. bulgaricus*, the structure of the skimmed fermented milk was fine-grained, with clear boundaries and visible cells (Figure 3A). The structure of the *L. plantarum* AG10 milk gel was characterized by the presence of a film between the protein particles, resulting in a loss of clear boundaries and more monolithic structure (Figure 3B). In the whole milk samples, fat particles were dispersed as small spheres or droplets, and fewer voids were observed (Figure 3C). As in the skimmed milk, clear *L. bulgaricus* cells were visible; particles of milk protein were bound into micelles by small clear particles (Figure 3D). The fat milk fermentation by *L. plantarum* AG10 led to the formation of large aggregates with less distinct inter-particle spaces, bound into a single conglomerate with some pores.

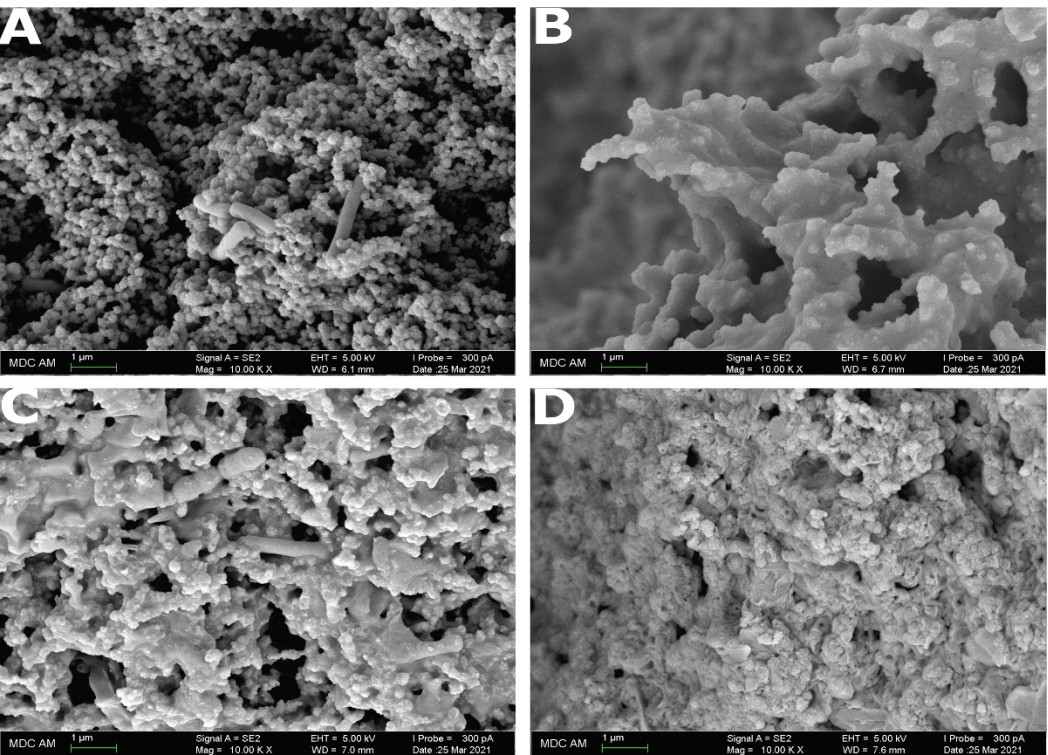

**Figure 3.** SEM images of fermented milk after 7 days of storage. (**A**) Skimmed milk fermented by *L. bulgaricus*; (**B**) skimmed milk fermented by L. *plantarum* AG10; (**C**) whole milk fermented by *L. bulgaricus*; (**D**) whole milk fermented by *L. plantarum* AG10.

### 3.5. Antioxidant Properties

The antioxidant properties of fermented milk samples represent the antioxidant potential of the strain as a protector of milk fat. The antioxidant potential was assessed in the Ferric reducing antioxidant power (FRAP) assay. The FRAPs of *L. bulgaricus* and *L. plantarum* AG10-fermented milk and its whey were similar on the first day after fermentation. Further, the antioxidant potential considerably increased during storage on the 14th day in *L. plantarum* AG10-fermented whole milk and on the 7th day for skimmed milk (Figure 4). Of note, the FRAP activity of the PFEs of both types of milk fermented by *L. plantarum* AG10 was superior from the first day of storage in comparison with that of *L. bulgaricus*.

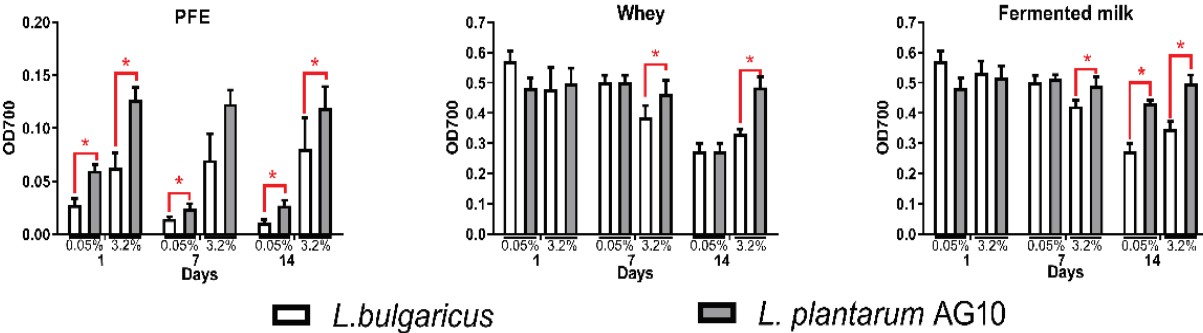

**Figure 4.** Antioxidant potentials of the protein-free extract (PFE), whey, and whole fermented milk obtained with either *L. bulgaricus* (shown in white) or *L. plantarum* AG10 (shown in gray) measured by the Ferric reducing antioxidant power (FRAP) assay after 1, 7, and 14 days of storage. Asterisks show a significant difference at $p < 0.05$. The % of initial fat in the milk is shown under the *x*-axis.

The study of the RSA of the fermented milk itself, whey, and PFE did not reveal any difference between the samples obtained with either *L. plantarum* AG10 or *L. bulgaricus* (Figure 5). Of note, a higher antioxidant activity was observed for the whole product.

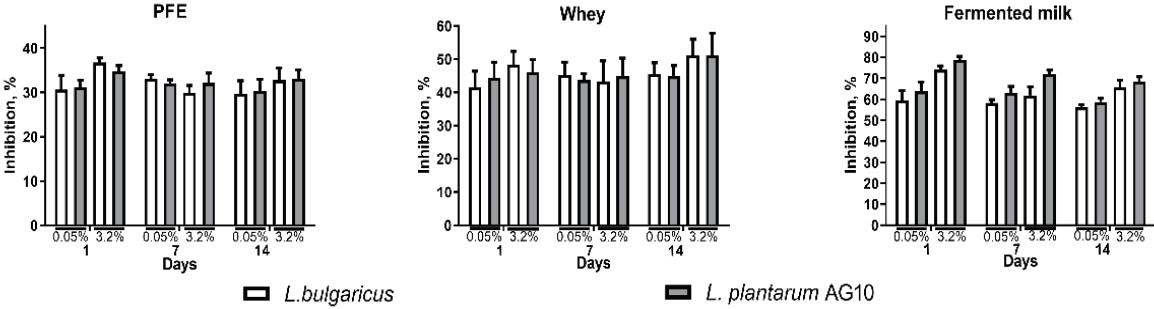

**Figure 5.** Radical-scavenging ability of protein-free extracts (PFE), whey, and whole fermented milk obtained with either *L. bulgaricus* (shown in white) or *L. plantarum* AG10 (shown in gray) measured in radical-scavenging ability (RSA) with the DPPH assay after 1, 7, and 14 days of storage. Asterisks show a significant difference at $p < 0.05$. The % of initial fat in the milk is shown under the *x*-axis.

Finally, the peroxide number, FFA, and TBARS were lower throughout the storage time in the milk fermented by *L. plantarum* AG10, suggesting the presence of metabolites with antioxidant properties (Figure 6).

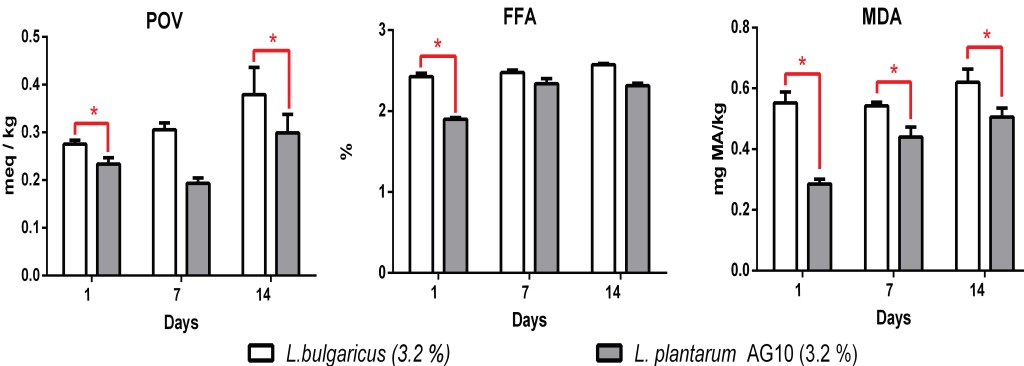

**Figure 6.** Peroxide value [55], free fatty acid value [2], and thiobarbituric acid-reactive substance (TBARS) of whole milk fermented with either *L. bulgaricus* (shown in white) or *L. plantarum* AG10 (shown in gray) after 1, 7, and 14 days of storage. Asterisks show a significant difference at $p < 0.05$. The % of initial fat in the milk is shown under the *x*-axis.

### 3.6. Viability of L. plantarum AG10 and L. bulgaricus during Storage

In the fermented milk, the number of *L. plantarum* AG10 CFUs was one order of magnitude higher in comparison with *L. bulgaricus* (Figure 7). Moreover, the number of *L. bulgaricus* CFUs decreased faster during storage, being 0.76 times less after 14 days in comparison with the initial level, while the *L. plantarum* AG10 cells decreased by a factor of 0.18. Of note, in the whole milk, the number of CFUs was higher in comparison with the skimmed milk, which may have been because of the protective effect of the fat.

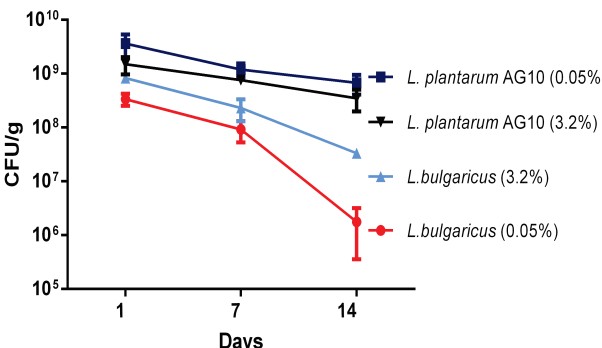

**Figure 7.** Amount of viable LAB in skimmed and whole milk fermented with either *L. bulgaricus* or *L. plantarum* AG10 measured by direct CFUs count after 1, 7, and 14 days of storage. The % of initial fat in the milk is shown.

### 4. Discussion

Milk fermentation by LAB serves to both enhance its nutritional value and provide probiotic strains for the correction of the intestinal microflora. Lactic acid bacteria, the indigenous microflora of naturally fermented foods, appear to be a promising source for new strains for dairy production as the main or minor starter cultures [38].

Recently, a series of LAB strains isolated from silage [43] were described, among which *Lactiplantibacillus plantarum* AG10 exhibited promising properties. Here, we show that *L. plantarum* AG10 ferments both skimmed and whole milk, resulting in a high-quality product. *L. plantarum* belongs to the NSLAB group, and its ability to acidify milk is not usually tested, but a number of studies have shown the ability of *L. plantarum* to cause the acidification of skimmed milk. For example, a number of strains from Feta cheese could lower the pH by 0.92 units [56]. It has been reported in several investigations that *L. plantarum* strains are able to grow and acidify enriched raw milk, but only in the presence of

a number of amino acids as an enrichment agent. Other authors have shown the ability of *L. plantarum* [57] to ferment milk, but at a lower rate compared with LAB starters. *L. plantarum* AG10 strain exhibited the ability to ferment both whole and skimmed milk, although at a lower rate than *L. bulgaricus* (Table 1), which resulted in a higher pH and lower TTA of the product, despite the higher carbohydrate consumption, as evidenced by the lower amount of residual total carbohydrates and dextrose in milk (Table 2). Apparently, this could be a consequence of the heterofermentative metabolism of *L. plantarum*, allowing the production of various metabolites [58], in contrast to heterofermentative *L. bulgaricus*, which converts sugars to lactic acid and thus provides higher acidification. While an obvious limitation of the low pH of fermented milk seems to be the decreased antimicrobial activity (Table 1), the lower acidity could be beneficial for people with increased gastric acid formation.

In contrast, *L. plantarum* AG10 produces 1.5-fold more EPS during skimmed milk fermentation (Figure 1), which results in the higher viscosity of the final product from skimmed milk and provides higher syneresis for fermented whole milk (Figure 2), assuming higher immune-regulating properties [31] and the texture of dairy products. The latter was also confirmed by SEM and the structural properties of the fermented milk. Thus, the *L. plantarum* AG10-fermented milk was characterized by higher density on microphotographs (Figure 3), with increased firmness and cohesiveness of the product (Table 2). In turn, the increase in the cohesiveness may indicate that more of the gel bonds that were destroyed during squeezing were restored after the cessation of the stress [59]. The *L. plantarum* AG10-fermented samples were correspondingly better at restoring these bonds in the gel, which was facilitated by EPS. Such a positive effect on the improvement of texture characteristics of yogurt by *L. plantarum* being added together with the starter cultures has already been shown previously for other *L. plantarum* isolates [60].

Besides the others, the antioxidant properties of LAB-fermented products have recently been extensively investigated [28,61], and higher antioxidant properties have been reported for *L. plantarum* strains [28]. While in the DPPH tests both *L. bulgaricus* and *L. plantarum* AG10-fermented milk exhibited a similar ability to bind free radicals, significantly lower POV, FFA, and TBARS were observed for *L. plantarum* AG10-fermented milk (Figure 6). This could be a consequence of either the intensive production of EPS (Figure 2), which have been reported as an antioxidant agent [30], or another metabolite produced by LAB (amino acids, uric acid, vitamins C, E, D, and A, β-carotene, enzymes SOD, catalase, and glutathione peroxidase) [32,61]. Nevertheless, the higher antioxidant activity suggests that the *L. plantarum* AG10-fermented dairy product would retain the best conditions for milk fat during storage.

## 5. Conclusions

Generally, our data show that the novel strain *L. plantarum* AG10 allows obtaining fermented skimmed and whole milk in some technological and textural properties superior to those of the product obtained by using *L. bulgaricus*. Thus, *L. plantarum* AG10 yields dense, viscous milk clots with lower acidity than *L. bulgaricus*-fermented milk, which could in turn be beneficial for people with increased gastric acid formation. In contrast, the milk fermented with *L. plantarum* AG10 after 14 days of storage retained a higher number of viable cells as well as demonstrated attractive textural properties caused by an increased EPS content and subsequent higher viscosity. Moreover, the *L. plantarum* AG10-fermented milk exhibited increased antioxidant potential and radical-scavenging activity in comparison with *L. bulgaricus*-fermented milk. Overall, *L. plantarum* AG10 seems a promising supplement for co-starter cultures in whole and skimmed dairy production.



**Author Contributions:** This study was made possible through the collaboration of all authors. Conceptualization, E.N. and A.K.; methodology, E.N. and A.K.; software, M.N.Y. and A.K; validation, A.K.; formal analysis, A.V., T.P. and A.E.; investigation, A.V., T.P. and A.E.; resources, A.K., E.N. and M.N.Y.; data curation, A.V., T.P. and A.E; writing—original draft preparation, E.N., M.N.Y. and A.K.; writing—review and editing, E.N., M.N.Y. and A.K.; visualization, A.K. and M.N.Y.; supervision, A.K. and E.N.; project administration, E.N. and A.K.; funding acquisition, A.K. and E.N. All authors have read and agreed to the published version of the manuscript.

**Funding:** The study was funded by RFBR (for EN, project number 20-016-00025) and supported by the Kazan Federal University Strategic Academic Leadership Program.

**Institutional Review Board Statement:** Not applicable.

**Informed Consent Statement:** Not applicable.

**Data Availability Statement:** All data are available in the manuscript file.

**Conflicts of Interest:** The authors declare no conflict of interest.

## Abbreviations

| | |
|---|---|
| LAB | Lactic acid bacteria |
| NSLAB | Non-specialized lactic acid bacteria |
| TTA | Total titratable acidity |
| FRAP | Ferric reducing antioxidant power |
| RSA | Radical-scavenging ability |
| WHC | Water-holding capacity |
| DPPH | 2,2-di-phenyl-1-picrylhydrazyl |
| PFE | Protein-free extract |
| TBARS | Thiobarbituric acid reactive substance |
| TBA | 2-thiobarbituric acid |
| MA | Malonaldehyde |
| FFA | Free fatty acid |
| POV | Peroxide value |

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
