# Peer review of "Textural and Functional Properties of Skimmed and Whole Milk Fermented by Novel Lactiplantibacillus plantarum AG10 Strain Isolated from Silage"

_fermentation, doi:10.3390/fermentation8060290_

Round 1
Reviewer 1 Report
The work "Textural and functional properties of skimmed milk and fat milk fermented by novel Lactiplantibacillus plantarum AG10 strain isolated from silage" has an interesting subject of the study. Still, there are issues that raise the reviewer's concern.
1. Terminology needs a thorough revision. We haven't fat milk, etc.
2. Fermentation process needs more detail (please explain the chosen 8 hours).
3. I wonder why peroxide value and free fatty acids have been analysed.
4. Analytical methods must be completed with validation aspects (standard buffers reliable, reference materials, accuracy to the second decimal of pH and other parameters).
5. Information about the data processing is missing in the Materials and methods.
6. A confusing information is given in Table 1, especially the total carbohydrates and pH data. Samples with L.bulgaricus are over-fermented, and samples with L.planturum AG10 are not completely fermented. For casein, the isoelectric point is approximately 4.4-4.6 and it is a pH value at which casein is precipitated. Precipitation influences casein gell firmness, and other structural data.
7. Discussion part has been revised and supplemented.
8. Conclusion part needs a polish (some specific comments are added in the manuscript).
9. Some comments are given in the manuscript (attached).

Author Response
Dear Editor,
thank you for providing us with three competent Reviewer reports on our ms. We would like to thank the Reviewers for taking their time to read our ms carefully and provide helpful suggestions. In the view of the constructive criticism by the Reviewers, we have revised the ms considerably. As you can see, we have revised the ms accordingly all suggestions raised by reviewers and added all requested information where applicable.
In the following, we response to particular concerns raised by the Reviewers in a step-by-step manner.
Reviewer #1:
The work "Textural and functional properties of skimmed milk and fat milk fermented by novel Lactiplantibacillus plantarum AG10 strain isolated from silage" has an interesting subject of the study. Still, there are issues that raise the reviewer's concern.
Reviewer comment:
- Terminology needs a thorough revision. We haven't fat milk, etc.
Author’s response:
Terminology has been corrected throughout the manuscript as suggested
Reviewer comment:
- Fermentation process needs more detail (please explain the chosen 8 hours).
Author’s response:
We fermented milk for 8 hours as this is the standard, most used milk fermentation time for yogurt production under factory conditions. It was important for us to evaluate the acid-synthesizing potential of L.plantarum AG10 during milk fermentation in comparison with L.bulgaricus. Therefore, we did not increase the time of fermentation for L.plantarum to values of pH=4.5. In this work, we did not aim to obtain a quality yogurt-like product based on the strains used (for that the conditions should be optimized), our aim was to identify and evaluate differences between L.plantarum AG10 and L.bulgaricus and assess the possibility of further use of this strain already when making a starter based on both L.bulgaricus and L.plantarum AG10.
Reviewer comment:
- I wonder why peroxide value and free fatty acids have been analysed.
Author’s response:
We analyzed peroxide value and free fatty acids indicators of whole milk to evaluate the antioxidant potential of the strain as a protector of milk fat. In the future, we plan to use L.plantarum strain in the composition of starters in the production of dairy products with high fat level (cheese, fermented cream, fat and whole yogurt and others).
Reviewer comment:
- Analytical methods must be completed with validation aspects (standard buffers reliable, reference materials, accuracy to the second decimal of pH and other parameters).
Author’s response:
All procedures were made accordingly the published description, the references are given in Methods section. We tried to add the information were applicable.
Reviewer comment:
- Information about the data processing is missing in the Materials and methods.
Author’s response:
The new section has been added
2.15 All experiments were carried out in triplicate. Five replicates were performed for DPPH analysis. The results were analyzed with non-parametric Kruskal–Wallis by GraphPad Prism 6.0 Software. Differences were considered significant at p < 0.05.
Reviewer comment:
- A confusing information is given in Table 1, especially the total carbohydrates and pH data. Samples with L.bulgaricus are over-fermented, and samples with L.planturum AG10 are not completely fermented. For casein, the isoelectric point is approximately 4.4-4.6 and it is a pH value at which casein is precipitated. Precipitation influences casein gell firmness, and other structural data.
Author’s response:
We agree with the Reviewer regarding effects of pH and low pH for L.planturum AG10 fermented milk. Nevertheless, as we answered to Q2, our research is primarily aimed at assessing the potential of L.planturum AG10. Further, a more significant indicator for assessing the quality of dairy products is the titratable acidity, in our country the level of acidity of the beverage must be between 70-140 °T, and this level is achieved in our experimental desing. In the long term, this gives us the possibility to use the strain as a co-culture to obtain another flavor and aroma of the final product, prevent excessive acidification and to suspend post-acidification processes during storage.
Furthermore, apparently, this could be a consequence of heterofermentative metabolism of L. plantarum allowing produce various metabolites in contrast to heterofermentative L. bulgaricus that converts sugars to lactic acid and thus provides higher acidification.
Reviewer comment:
- Discussion part has been revised and supplemented.
Author’s response:
The section has been revised
Reviewer comment:
- Conclusion part needs a polish (some specific comments are added in the manuscript).
Author’s response:
The section has been revised
Reviewer comment:
- Some comments are given in the manuscript (attached).
Author’s response:
We added answers to comments
Dr. Elena Nikitina and Dr. Airat Kayumov,
For all authors

Reviewer 2 Report
This article illustrates a work about skimmed and fat milk fermented by a novel Lactiplantibacillus plantarum AG10 strain. The authors tried to confirm the potential of this strain as a starter for the fermented milk compared with Lactobacillus delbrueckii subs. bulgaricus. The study included a large number of testing experiments on the skimmed and fat fermented milk during the storage of 14 d, such as texture, syneresis, WHC, EPS, FRAP, RSA, CFU, etc. However, this article needs to be revised and improved carefully before publication. The comments are as following:
1. The language needs to be carefully checked and revised. There are many spelling mistakes (Line 370, 423 and Table 1, etc) and long sentences which are incorrect, confusing and unclear in the article.
2. Making a list of abbreviations is suggested.
3. Abstract should be rewritten by indicating important results with specific data and significance.
4. Introduction
1) Please reorganize the bibliographic ideas. Please focus on the properties tested in this article such as texture and antioxidant activity, and summarize the recent research in such field.
2) Line 71-82. The last section of the introduction is usually important and should include the research purpose and work content of the article, not the results.
5. Material and methods
1) Please reorder each experiment according to the order in “3. Results”.
2) 2.1. Strains and milk fermentation: Please indicate the storage condition of the strain and the initial inoculum amount for the fermentation.
3) The statistically analysis method must be included.
6. Results
1) Tables: Please show more detail information in the captions to be easily understood. And the significant digits need to be adjusted according to measurement accuracies.
2) Due to its measurement principle and calculation method, the syneresis and WHC are frequently negatively correlated, that is, samples with high syneresis usually have low WHC. This principle does not match the results of present study. Please verify this result.
3) FRAP and RSA: Please express with IC50 and positive control compound, such as trolox.
4) Fig 3: Please show the storage time of the tested samples.
7. Discussions
1) Line 378-379: This is an incorrect description. LAB also includes L. plantarum.
8. Conclusions
Line 417: Please avoid emotive words such as "good" and used objective ones.
9. References
Too many references were cited. Some unimportant references can be deleted to match the requirement of the journal.
Author Response
Dear Editor,
thank you for providing us with three competent Reviewer reports on our ms. We would like to thank the Reviewers for taking their time to read our ms carefully and provide helpful suggestions. In the view of the constructive criticism by the Reviewers, we have revised the ms considerably. As you can see, we have revised the ms accordingly all suggestions raised by reviewers and added all requested information where applicable.
In the following, we response to particular concerns raised by the Reviewers in a step-by-step manner.
Reviewer #2:
This article illustrates a work about skimmed and fat milk fermented by a novel Lactiplantibacillus plantarum AG10 strain. The authors tried to confirm the potential of this strain as a starter for the fermented milk compared with Lactobacillus delbrueckii subs. bulgaricus. The study included a large number of testing experiments on the skimmed and fat fermented milk during the storage of 14 d, such as texture, syneresis, WHC, EPS, FRAP, RSA, CFU, etc. However, this article needs to be revised and improved carefully before publication. The comments are as following:
Reviewer comment:
The language needs to be carefully checked and revised. There are many spelling mistakes (Line 370, 423 and Table 1, etc) and long sentences which are incorrect, confusing and unclear in the article.
Author’s response:
The language has been corrected
Reviewer comment:
Making a list of abbreviations is suggested.
Author’s response:
List of abbreviations was added
Reviewer comment:
Abstract should be rewritten by indicating important results with specific data and significance.
Author’s response:
The section has been revised
Reviewer comment:
Introduction
Please reorganize the bibliographic ideas. Please focus on the properties tested in this article such as texture and antioxidant activity, and summarize the recent research in such field.
Author’s response:
The section has been revised
Reviewer comment:
Line 71-82. The last section of the introduction is usually important and should include the research purpose and work content of the article, not the results.
Author’s response:
The part has been revised as suggested
Reviewer comment:
Please reorder each experiment according to the order in “3. Results”.
Author’s response:
The section has been revised
Reviewer comment:
2.1. Strains and milk fermentation: Please indicate the storage condition of the strain and the initial inoculum amount for the fermentation.
Author’s response:
The section has been revised
Reviewer comment:
The statistically analysis method must be included.
Author’s response:
The new section has been added
2.15 All experiments were carried out in triplicate. Five replicates were performed for DPPH analysis. The results were analyzed with non-parametric Kruskal–Wallis by GraphPad Prism 6.0 Software. Differences were considered significant at p < 0.05.
Reviewer comment:
Results
Tables: Please show more detail information in the captions to be easily understood. And the significant digits need to be adjusted according to measurement accuracies.
Author’s response:
Tables 1 and 2 were corrected
Reviewer comment:
Due to its measurement principle and calculation method, the syneresis and WHC are frequently negatively correlated, that is, samples with high syneresis usually have low WHC. This principle does not match the results of present study. Please verify this result.
Author’s response:
We have revised data regarding data description. Since we have almost no difference on 1st and 7th day, the correlation is not visible. Moreover, when preparing the samples for WHS and syneresis determination, various centrifugation speeds are used, that also affects the correleation.
Reviewer comment:
FRAP and RSA: Please express with IC50 and positive control compound, such as trolox.
Author’s response:
For regret, we have no enough data to calculate IC50 and did not positive controls.
Reviewer comment:
Fig 3: Please show the storage time of the tested samples.
Author’s response:
The information is added
Reviewer comment:
Discussions
Line 378-379: This is an incorrect description. LAB also includes L. plantarum.
Author’s response:
Is corrected
Reviewer comment:
Conclusions
Line 417: Please avoid emotive words such as "good" and used objective ones.
Author’s response:
The section has been revised
Reviewer comment:
References
Too many references were cited. Some unimportant references can be deleted to match the requirement of the journal.
Author’s response:
The reference list has been revised
Dr. Elena Nikitina and Dr. Airat Kayumov,
For all authors
Reviewer 3 Report
#1 Over the past few years Texture Profile Analysis testing has been the cause of much concern. In general, TPA is a very popular method of testing, as it provides very quick calculation of parameters which are 'believed to correlate with sensory analysis'. The following is a set of points to consider when choosing TPA as your test procedure:
#a Size of Compression Probe versus Sample When the probe is larger than the sample, the forces registered are largely due to uniaxial compression. However, when the opposite is true, the forces derive largely from puncture, a combination of compression and shear. Various papers throughout the decades of using TPA have reported the use of probes both larger and smaller than the test samples. Early papers on TPA report the use of puncture probes, but in 1968 Prof. Malcolm Bourne was the first to adopt true uniaxial compression to perform TPA tests. Generally speaking, most recent work done on TPA uses compression probes of the same size as or larger than the sample size, so that the forces registered in such TPA tests are largely due to uniaxial compression forces and the whole of the sample piece is tested. Unfortunately, the authors did not provide any information about size of the samples. Therefore, if the probe was smaller then the sample this shortcoming must be addressed and influence on the obtained results explained.
#b Extent of Deformation Another area of abuse is the degree of compression. The original TPA work used 80% strain. Most of Dr. Bourne's TPA research was conducted at 90% strain. The premise was that most foods should be chewed very fully in order to successively break up the mass until it is acceptable to swallow. If breaking up the food until it is palatable to swallow is the test objective, then by all means test products using strains approximating 66% to 80%. However, the authors did not provid any info on the compression. They must explain the basis for such unusual decision?
At the end we would suggest the authors to exclude TPA results completly and provide olfactory sensory analysis ba the trained panel. This important peace of research is missing anyhow.
Author Response
Dear Editor,
thank you for providing us with three competent Reviewer reports on our ms. We would like to thank the Reviewers for taking their time to read our ms carefully and provide helpful suggestions. In the view of the constructive criticism by the Reviewers, we have revised the ms considerably. As you can see, we have revised the ms accordingly all suggestions raised by reviewers and added all requested information where applicable.
In the following, we response to particular concerns raised by the Reviewers in a step-by-step manner.
Reviewer #3:
#1 Over the past few years Texture Profile Analysis testing has been the cause of much concern. In general, TPA is a very popular method of testing, as it provides very quick calculation of parameters which are 'believed to correlate with sensory analysis'. The following is a set of points to consider when choosing TPA as your test procedure:
Author’s response:
Text was added:
Fermented milk was tested in a chemical beaker with a diameter of 50 mm, the volume of the test sample was 50 ml, the height of the sample was 25 mm.
Reviewer comment:
#a Size of Compression Probe versus Sample When the probe is larger than the sample, the forces registered are largely due to uniaxial compression. However, when the opposite is true, the forces derive largely from puncture, a combination of compression and shear. Various papers throughout the decades of using TPA have reported the use of probes both larger and smaller than the test samples. Early papers on TPA report the use of puncture probes, but in 1968 Prof. Malcolm Bourne was the first to adopt true uniaxial compression to perform TPA tests. Generally speaking, most recent work done on TPA uses compression probes of the same size as or larger than the sample size, so that the forces registered in such TPA tests are largely due to uniaxial compression forces and the whole of the sample piece is tested. Unfortunately, the authors did not provide any information about size of the samples. Therefore, if the probe was smaller then the sample this shortcoming must be addressed and influence on the obtained results explained.
Author’s response:
The basic methodology for measuring texture was uses dy Najgebauer-Lejko, D., et al., Changes in the viscosity, textural properties, and water status in yogurt gel upon supplementation with green and Puerh teas. Journal of Dairy Science, 2020. 103(12): p. 11039-11049.DOI: 10.3168/jds.2020-19032, with modification.
Reviewer comment:
#b Extent of Deformation Another area of abuse is the degree of compression. The original TPA work used 80% strain. Most of Dr. Bourne's TPA research was conducted at 90% strain. The premise was that most foods should be chewed very fully in order to successively break up the mass until it is acceptable to swallow. If breaking up the food until it is palatable to swallow is the test objective, then by all means test products using strains approximating 66% to 80%. However, the authors did not provid any info on the compression. They must explain the basis for such unusual decision?
Author’s response:
The aim of the measurement was to compare and identify differences in the textural properties of the gels formed by different strains.
The texture indicators were calculated according to the article
https://www.researchgate.net/publication/316093466_On_the_texture_profile_analysis_test
Reviewer comment:
At the end we would suggest the authors to exclude TPA results completly and provide olfactory sensory analysis ba the trained panel. This important peace of research is missing anyhow.
Author’s response:
We did not measure sensory characteristics since there is no finished fermented dairy product in the work. It should be taken into account that the work shows that the use of L. plantarum in a monoculture is not reasonable, because the rate of lactic acid accumulation in L. plantarum is less than that of bacillus. It is not economically advantageous. In the following studies, we plan to present the results of co-culture of L. plantarum with yogurt starter to produce probiotic yogurt.
Dr. Elena Nikitina and Dr. Airat Kayumov,
For all authors
Round 2
Reviewer 1 Report
I haven't objections or comments. Thanks for the work.
Reviewer 3 Report
The authors have successfully addressed all the issues raised by the reviewers. The manuscript can be published as is.